# Afrotropical sand fly-host plant relationships in a leishmaniasis endemic area, Kenya

Iman B. Hassaballa[1,2], Catherine L. Sole[2], Xavier Cheseto[1], Baldwyn Torto[1,2], David P. Tchouassi[1]*

1 International Centre of Insect Physiology and Ecology, Nairobi, Kenya, 2 Department of Zoology and Entomology, University of Pretoria, Pretoria, South Africa

* dtchouassi@icipe.org

## Abstract

The bioecology of phlebotomine sand flies is intimately linked to the utilization of environmental resources including plant feeding. However, plant feeding behavior of sand flies remains largely understudied for Afrotropical species. Here, using a combination of biochemical, molecular, and chemical approaches, we decipher specific plant-feeding associations in field-collected sand flies from a dry ecology endemic for leishmaniasis in Kenya. Cold-anthrone test indicative of recent plant feeding showed that fructose positivity rates were similar in both sand fly sexes and between those sampled indoors and outdoors. Analysis of derived sequences of the ribulose-1,5-bisphosphate carboxylase large subunit gene (rbcL) from fructose-positive specimens implicated mainly Acacia plants in the family Fabaceae (73%) as those readily foraged on by both sexes of *Phlebotomus* and *Sergentomyia*. Chemical analysis by high performance liquid chromatography detected fructose as the most common sugar in sand flies and leaves of selected plant species in the Fabaceae family. Analysis of similarities (ANOSIM) of the headspace volatile profiles of selected Fabaceae plants identified benzyl alcohol, (*Z*)-linalool oxide, (*E*)-*β*-ocimene, *p*-cymene, *p*-cresol, and *m*-cresol, as discriminating compounds between the plant volatiles. These results indicate selective sand fly plant feeding and suggest that the discriminating volatile organic compounds could be exploited in attractive toxic sugar- and odor- bait technologies control strategies.

## Author summary

Plant feeding as an essential resource of sand flies, primary vectors of *Leishmania* parasites, is largely understudied for Afrotropical species. Here, we combined field ecology, biochemical, molecular and chemical approaches, to decipher plant feeding associations in field-collected sand flies from a dry ecology endemic for leishmaniasis in Kenya revealing i) similar rates of plant feeding among sand fly sexes sampled from indoor and outdoor environments, ii) Acacia plants in the family Fabaceae as those readily foraged on by sand fly species in *Phlebotomus* and *Sergentomyia*, iii) fructose as the common sugar in sand flies and leaves of selected plant species in the Fabaceae family, iv) compounds

**Data Availability Statement:** All relevant data are within the paper and its Supporting Information.

**Funding:** IBH was supported by a German Academic Exchange Service (DAAD) In-Region Postgraduate Scholarship (Grant number:

91672086). This study was partly supported by the project, Combatting Arthropod Pests for better Health, Food and Climate Resilience (Project number: RAF-3058 KEN-18/0005) funded by Norwegian Agency for Development Cooperation (Norad). We also acknowledge the financial support for this research by the following organizations and agencies: UK's Foreign, Commonwealth & Development Office (FCDO), the Swedish International Development Cooperation Agency (Sida), the Swiss Agency for Development and Cooperation (SDC), the Federal Democratic Republic of Ethiopia and the Ministry of Higher Education, Science and Technology, Kenya. The views expressed herein do not necessarily reflect the official opinion of the donors. The funders had no role in study design, data collection and analysis, decision to publish, or preparation of the manuscript.

**Competing interests:** The authors have declared that no competing interests exist.

namely benzyl alcohol, (Z)-linalool oxide, (E)-β-ocimene, p-cymene, p-cresol, and m-cresol, as discriminating volatile organic compounds between volatiles of selected Fabaceae plants. The findings indicate selective sand fly plant feeding and suggest that the discriminating volatile organic compounds could be exploited in attractive toxic sugar- and odorbait technologies for sand fly control.

## Introduction

The sand fly-borne disease leishmaniasis, constitutes a public health problem in eastern Africa including Sudan, Ethiopia, Uganda, Kenya, and Somalia [1]. Leishmaniasis ranks among the priority list of zoonotic diseases for control in Kenya [2] where an estimated 4000 human cases of the visceral form (VL) of the disease occurs, with about 5 million at risk of infection [3]. Current control of leishmaniasis relies on prompt diagnosis and chemotherapeutic treatment. However, these approaches have not prevented the spread of the disease across Kenya. Visceral leishmaniasis has expanded in geographic range with frequent recent outbreaks as the cutaneous form (CL) of the disease [3]. New tools are therefore urgently needed with emphasis on disease prevention through effective management and control of the sand fly vectors.

Control of leishmaniasis by targeting the vectors can be achieved through an understanding of sand fly behavior, with prominent amongst them being plant feeding [4]. Plants serve as the primary source of energy for sand flies to support various biological functions including survival, fecundity, and dispersal. Male sand flies feed exclusively on plants. Females obligately rely on plant derived-diets only sporadically feeding on blood [5]. As such, plant feeding offers an attractive target for developing surveillance and control strategies to limit human pathogen exposure risk. Examples include attractive toxic sugar bait (ATSB) [6–9], an 'attract and kill' strategy whereby a sugar solution as feeding substrate is laced with oral low risk toxins either as bait stations or applied on host plants [10,11]. Potential transmission blocking agents could be envisaged from the understanding of plants fed upon by sand flies in nature. This essential behavior exposes them to a range of plant produced metabolites which may affect their survival but also their vector competence to pathogens. Schlein and Jacobson [5] demonstrated that feeding on certain plants by the sand fly *Phlebotomus papatasi* (vector of CL) modulated the outcome of infection with the parasite, *Leishmania major*. However, effective implementation of these intervention strategies requires precise information on plant feeding sources.

The use of sensitive molecular techniques such as polymerase chain reaction (PCR) and sequencing has improved our understanding of specific forage associations of sand flies and other arthropod disease vectors [12,13]. Using these approaches, Lima et al [14] demonstrated feeding preference by the Neotropical sand fly species *Lutzomyia longipalpis* on plants in the Fabaceae family. Recently, several leishmaniasis sand fly vectors were shown to prefer the plant *Cannabis sativa* for sugar feeding [15]. The specific plants that Afrotropical species of sand flies forage in nature are poorly understood. Specific sand fly-plant associations are reported in literature; for example, the occurrence of the sand fly vector *Phlebotomus orientalis* in habitats with *Acacia seyal* and *Balanites aegyptiaca* vegetation [16]. Nonetheless, understanding the ecologic functions underlying such relationships have not been fully described.

The objective of this study was to document the degree to which diverse wild-caught sand fly species including *Phlebotomus* vectors of leishmaniasis utilize plant feeding sugar resources in the natural setting and explore the nutritional and semiochemical basis for host plant selection for feeding.

## Materials and methods

### Ethics statement

Approval for the study was sought from the Scientific Ethics and Review Committee of the Kenya Medical Research Institute (SERU-KEMRI) (Protocol number: 3312). In addition, verbal consent was obtained from the chief of Rabai village and heads of households selected for sampling sand flies from inside and outside houses.

### Sample collection

Adult phlebotomine sand flies were collected in Rabai (0.45866˚ N, 35.9889˚ E), a rural community endemic for VL and CL located in the Marigat sub-County, Baringo County, Kenya (Fig 1). The area is a dry ecological zone characterized by numerous large termite mounds and sparse vegetation mostly over-grazed by livestock [17,18]. The vegetation mainly consists of thorny and *Commiphora* bushes or Kanniedood, Acacia trees, *Cactus* trees, and the invasive plant species *Prosopis juliflora* [18]. Sand flies were surveyed using Centers for Disease Control (CDC) miniature light traps (Model 512, John Hock Co., Gainesville, Florida, USA) in December 2018. Three - four traps were randomly deployed to collect night-active sand flies daily for eight consecutive days from three habitat types: houses indoors and outdoors, around termite mounds and animal sheds. Sand flies were sorted, immediately frozen in liquid nitrogen and

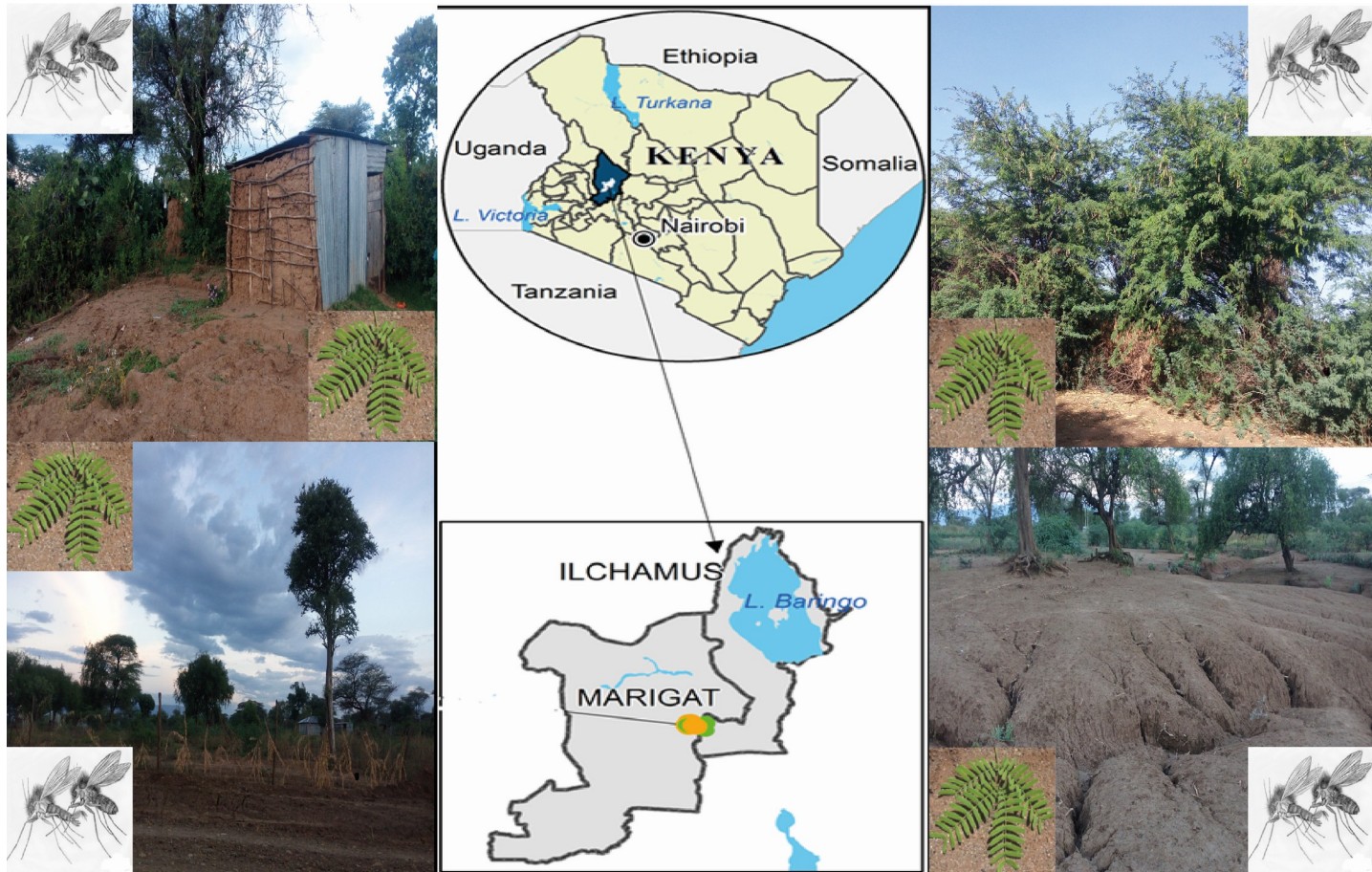

**Fig 1. Map of the study site showing the environment where sand flies and plants were sampled from Rabai village in Baringo County, Kenya.**

transported to the laboratory at International Centre of Insect Physiology and Ecology (ICIPE) in Nairobi, where they were stored at –80˚C until further processing.

## Sample preparation and identification

To remove any plant debris from the body of the sand fly, individual specimens were rinsed in 0.5% bleach solution for 1 min, and then twice in double distilled water (ddH$_2$O) each lasting 30 s as described previously in Nyasembe et al [12]. Sand flies were morphologically identified to species based on dissection and observation of the external genitalia (for males) and the pharynx, cibarium and spermatheca (for females) using published keys [19,20]. The remaining part of the body (abdomen) containing the crop was processed for biochemical and molecular analyses.

## Evidence of recent plant feeding in the field sand flies

Each sand fly abdomen containing the crop was macerated in 50 μL absolute ethanol using a sterilized pestle. An aliquot (25 μL) of the sand fly homogenate from each individual was placed in the wells of a flat-bottomed 96-well microplate, and tested for presence of fructose as evidence of recent plant feeding by the cold anthrone test [21] with modifications by Matheson et al [22]. Briefly, 200 μL anthrone solution (0.15% anthrone (Sigma -Aldrich) w/v in 71.7% sulphuric acid) was added to each well containing the homogenate and incubated for 60 min at room temperature (25˚C). A change in color from yellow to green or blue was deemed indicative of a positive test [21].

## Plant DNA extraction from sand fly homogenate

DNA was extracted from the aliquot of homogenate (above) of fructose–positive samples using the ISOLATE II Plant DNA Kit (Bioline, London, UK) according to the manufacturer's instructions with a slight modification. This included extension of the incubation period with the lysis buffer PA1 and RNase by 4–6 hr and elution with Buffer PG by 10 min. The extracted DNA was stored at -20˚C until further processing.

## PCR amplification, purification, and sequencing

A PCR target of 450–660 bp fragment of the ribulose-1,5-bisphosphate carboxylase large sub-unit gene (rbcL) was used to identify plant DNA in sand flies, a widely used marker for plant barcoding studies. The primers used included rbcLaF (5'- ATGTCACCACAAACAGAGAC-TAAAGC-3') and rbcLaR (5'- GTAAAATCAAGTCCACCRCG -3') [23,24]. Polymerase chain reaction (PCR) MyTaq DNA Polymerase Kit (Bioline, London, UK) in a total of 10 μL reaction volume containing 10μM each of the forward and reverse primers, 0.0625 U MyTaq DNA polymerase, 5X My Taq reaction buffer and 1–2μL of DNA template (1–10 ng) was used for PCR using a Veriti 96-well Thermal Cycler (Singapore). Polymerase chain reaction thermal profile included initial denaturation at 94˚C for 4 min, followed by 35 cycles of 94˚C for 30 sec, 55˚C for 30 sec and 72˚C for 1 min, and final extension at 72˚C for 10 min. Amplicons were resolved on 1.5% agarose gel electrophoresis stained with ethidium bromide (Sigma-Aldrich, GmbH, Germany) against a 100 bp DNA ladder (HyperLadder, Bioline, London, UK). Polymerase chain reaction water served as negative controls in all PCRs and DNA extracted directly from the tissue of selected plants (described below) as positive controls. Polymerase chain reaction products were purified using the ISOLATE II PCR and Gel Kit (Bioline, London, UK) or Exo/SAP-IT Kit (Affymetrix Inc., USA) as per the manufacturers'

instructions. Purified products were outsourced for Sanger sequencing to generate both forward and reverse reads to Macrogen (Inc Europe Laboratory).

### DNA sequence analysis

The forward and reverse sequences were cleaned as described in Tchouassi et al. [25]. Briefly, they were visually inspected, aligned and edited using MEGA v. 7 [26]. Consensus sequences for each sample were aligned using ClustalW in MEGA using default parameters. Each sequence was compared to a reference sequence in the GenBank database using BLASTn with the search option 'Highly similar sequences'. Sequences were assigned to specific plant species based on matches >98% [27].

### Plant sample collection as a reference for identified sand fly host plants

The leaves of selected putatively identified plants based on sequence analysis were collected from the study site. The leaves were wrapped separately for each plant in newspaper in the field, labeled, and transported to the laboratory at ICIPE in Nairobi and dried at room temperature. 20 mg dry weight each of the grounded plants was similarly processed for DNA extraction, amplification and sequencing as described previously.

### Processing of plants and sand flies for sugar profiling

Leaves were sampled from selected Fabaceae plants including *Vachellia tortilis*, *Senegalia laeta*, *Vachellia nilotica* and *Prosopis juliflora* in the field which were identified as host plants for sand flies through sequence analysis (see Results section below). Also, two other Acacia plants *Senegalia senegal*, *Vachellia elatior*, present in the study locality but their DNA not detected in the sand flies were sampled. The plant leaves were harvested and transported in wrapped newspaper to the lab at ICIPE in Nairobi, Kenya. They were separately air-dried in the shade at room temperature and then ground to fine powder using an electric grinder (Retsch GmbH, Haan, Germany) and extracted following the method of Mokaya et al [28] with slight modifications. Briefly, 20 mg of each of the ground plant samples (leaves) was extracted in 1 ml (3:1 distilled deionized water: acetonitrile), vortexed for 10s, sonicated for 1 h, and centrifuged at 14,000 rpm for 5 min. The supernatant was filtered using qualitative filter paper (Whatman, circles, diam. 25 mm) and analyzed for sugars by High Performance Liquid Chromatography (HPLC) (described below). The plant materials were analyzed in triplicate, with each replicate collected from different plant samples. The abdomen containing the crop was dissected from wild caught sand flies and pooled (50/pool) by sex, extracted, and analyzed for sugars as described for the plants.

### Chemical analysis of sugars in identified plants and sand flies

For sugar detection, 10 μl of each plant and sand fly sample extract was analyzed by HPLC following previously established methods for sugars [28]. This was carried out on an Agilent HPLC system (Palo Alto, CA) equipped with a 1260 refractive index detector (RID) and photo diode array detector (wavelength set at 190−360 nm for UV and 360−700 nm for visible range). The column oven temperature was set at 30˚C with the following column parameters, 250mm × 4.6mm i.d., 5 μm, LC-NH2 column (Supelco, Bellefonte, PA, USA). The mobile phases consisted of an isocratic system water (A) and acetonitrile (B), (75:25 v/v). Samples were tested for presence of the following sugars: fructose, glucose, sucrose, galactose, maltose, xylose, trehalose and lactose, which were identified by comparing their retention times and co-injection with those of the authentic samples. Serial dilutions of the authentic sugar standards

(0.1–100 ng/μl) were analyzed also to generate linear calibration curves which gave coefficient of determinations of $R^2$; fructose (0.9999), glucose (0.9998) and sucrose (0.9999). These regression equations were used for the external quantification of the different sugars found in the plants and sand fly samples, respectively.

## Collection of volatiles from selected identified plants in the field

Headspace volatiles targeting the vegetative parts were collected from four of the identified host plants fed on by sand flies (~70% of samples analyzed) based on plant DNA detection namely, *V. tortilis*, *S. laeta*, *V. nilotica* and *P. juliflora* (see results section). This was carried out *in situ* in their natural habitats at the study site, using portable field pumps (Analytical Research System, Gainesville, Florida, USA). Odors from the four plants were collected by enclosing a vegetative part in an airtight oven bag (Reynolds, Richmond, VA, USA) (S1 Fig) by passing charcoal-filtered air at a flow rate of 350 ml/min on to two Super-Q adsorbents (30 mg, Analytical Research System, Gainesville, Florida, USA) for each replicate substrate. For all the plant species, volatiles were collected for 12 hr (06:00–18:00) during the day and 12 hr (18:00–06:00) at night and replicated three to four times for each plant species using a different plant. The same procedure was applied for the control, comprising volatiles from blank oven bags in the same habitat of the target plant. The Super-Q traps were each eluted with dichloromethane (DCM) (200 μl) (GC-grade, Sigma Aldrich, Gilling-ham, UK) and analyzed using coupled gas chromatography/mass spectrometry (GC-MS).

## Chemical analysis of volatiles from selected identified plants

Plant volatile extracts were analyzed using GC-MS. An aliquot (1 μl) of DCM volatile extract of each sample and a blank were injected into GC-MS in a splitless injection mode. The GC was equipped with an HP-5 column (30 m x 0.25 mm ID x 0.25 μm film thickness) with helium as the carrier gas at a flow rate of 1.2 ml/min. The oven temperature was held at 35°C for 5 min, then programmed to increase at 10°C/min to 280°C and was maintained at this temperature for 10.5 min. The mass selective detector was maintained at ion source temperature of 230°C and a quadrupole temperature of 180°C. Electron impact (EI) mass spectra was obtained at the acceleration energy of 70 eV. Fragment ions were analyzed over 40–550 *m/z* mass range in the full scan mode. The filament delay time was set at 3.3 min.

The compounds were identified by comparison of mass spectrometric data and retention times with those of authentic samples and reference spectra published by library–MS databases: Adams2, Chemecol and NIST (0.5, 0.8 and 11). Compounds present in controls were excluded from compositional profiles in each sample. Furthermore, identification of the VOCs in each plant was achieved based on their retention indices (RI) which were determined with reference to a homologous series of normal alkanes $C_8$-$C_{23}$ and calculated using the equation below as described by Van den Dool and Kratz [29] and comparison with published literature [30–32].

$$RIx = 100\ n_0 + 100\ (R_Tx - R_Tn_1)/(R_Tn_1 - R_Tn_0)$$

With:
x = the name of the target compound
$n_1$ = n-alkane $C_{n1}H_{2n1+2}$ directly eluting before x
$n_0$ = n-alkane $C_{n0}H_{2n0+2}$ directly eluting after x
$R_T$ = retention time
RI = retention index

## Chemicals

The chemicals used for HPLC including D-(+)-xylose, 99%, D-(+)-galactose 99%, D-(-)-fructose 99%, sucrose 99.5%, maltose 99%, D-(+)-glucose 99.5%, D-(+)- trehalose 99%, D-lactose 99.5%) and acetonitrile (ACN) (HPLC grade) were purchased from Sigma-Aldrich, St. Louis, Missouri, United States. Chemicals used in the GC-MS analysis including hexanal, heptanal, benzaldehyde, octanal, nonanal, decanal, 6-methyl-5-hepten-2-one, acetophenone, $\alpha$-pinene, $p$-cymene, sabinene, $\beta$-pinene, (E)-$\beta$-ocimene, (Z)-linalool oxide (furanoid), (Z)-linalool oxide (pyranoid), $\alpha$-cedrene, octanol, 1-octen-3-ol, phytol, benzyl alcohol, $m$-cresol, $p$-cresol, indole, methyl salicylate and standard n-alkanes solution were purchased from Sigma Aldrich. All the chemicals were >97% purity.

## Statistical analysis

The proportion of fructose positive sand flies sampled outdoor (termite mounds + animal shed) and indoor (houses), was subjected to a Chi- squared test to compare proportions. The amount of sugar type in the different plants and sand fly crop was expressed as mean ± standard error based on 3 replicate runs. Analysis of variance (ANOVA) was carried out to compare the amounts for each sugar detected between the plants and sand fly species followed by mean separation using Tukey's HSD (Honest Significant Test) after checking for normality using Shapiro-Wilk test. All analyses were performed in R version 3.6.3 [33] at 95% significance level. Similarity percentage (SIMPER) analysis was performed on peak areas of volatile compounds identified by GC-MS to determine the relative contribution of different compounds to the dissimilarity among volatiles of the different plants. The output was visualized using the non-metric multidimensional scaling approach. One-way analysis of similarities (ANOSIM) using Bray–Curtis dissimilarity matrix was performed to compare the chemical profiles of different plants volatiles using Past 3 free software [34]. The frequency of plant meal sources for each sand fly regardless of sex was classed into families and visualized using the heatmap, generated using Past 3 free software [34].

## Results

### Evidence of recent plant feeding in wild-caught sand fly species

Six hundred sand flies were tested for fructose by the cold anthrone test. These comprised 300 sampled indoors in houses and 300 outdoors from both termite mounds and animal sheds.

Overall, 38.7% (116/300) were fructose positive indoors compared to 35.7% (107/300) outdoors. The difference was not statistically significant ($\chi$2 = 0.36, $df$ = 1, $P$ = 0.5). Fructose positive rate of 44.7% (67/150) for females and 32.7% (49/150) for males did not differ significantly for collections indoors ($\chi$2 = 2.8, $df$ = 1, $P$ = 0.09). Similarly, for outdoor collections, 33.3% (50/150) males and 38% (57/150) females tested positive for fructose and the difference was not statistically significant ($\chi$2 = 0.46, $df$ = 1, $P$ = 0.5). The data by species are presented in Table 1.

### Plant DNA amplification from fructose-positive sand flies

A subset of fructose positive sand flies (n = 221) were processed by PCR of which 83 amplified for rbcL (Table 2). Forty of the rbcL amplicons were successfully sequenced (Table 2). Among unsuccessful sequenced specimens were those (n = 5) from which multiple DNA bands indicative of mixed plant meals (from different host plants) were detected. Polymerase chain reaction (PCR) success rate and sequencing varied between the different sand fly species analyzed. The

**Table 1. Sugar feeding status of wild-caught phelobotomine sand fly species in Rabai village during the dry season.**

| Sand fly species | Indoor (No. fructose positive (No. tested) | | Outdoor (No. fructose positive (No. tested) | |
|---|---|---|---|---|
| | Male | Female | Male | Female |
| *P. duboscqi* | 0(0) | 0(0) | 2(7) | 5(7) |
| *P. martini* | 0(5) | 1(5) | 7(45) | 4(45) |
| *S. antennata* | 13(98) | 24(98) | 1(25) | 6(25) |
| *S. africana africana* | 9(58) | 7(58) | 0(7) | 0(7) |
| *S. clydei* | 0(3) | 0(3) | 1(15) | 1(15) |
| *S. schwetzi* | 27(131) | 32(131) | 39(179) | 36(179) |
| *S. squamipleuris* | 0(5) | 3(5) | 0(22) | 5(22) |
| Total | 49(150) | 67(150) | 50(150) | 57(150) |

sequenced fragment sizes ranged between 450 and 660 bp. The data presented indicate plant DNA detection from single plant meals (one host plant) in individual samples.

## Afro-tropical sand flies preferentially feed on plants in the Fabaceae family

Analysis of the 40 derived rbcL sequences from the sand flies implicated 16 plant species as feeding sources belonging to 5 families *viz*: Fabaceae, Musaceae, Solanaceae, Poaceae, Lauraceae (Fig 2; S1 Table). The plants identified for males as candidate sugar sources belong to the families Fabaceae, Musaceae, Solanaceae and Lauraceae, whereas for females they were Fabaceae, Poaceae, and Musaceae. As such, the key plant families associated with both sexes included Fabaceae, Musaceae and Poaceae. Most of the plants identified were Acacias belonging to the family Fabaceae and represented by at least 4 species *V. tortilis*, *S. laeta*, *V. nilotica* and *Faidherbia albida* (S1 Table). Analysis of the plant species profiles of the Fabaceae showed that sand flies of both sexes had predominantly fed on *V. tortilis*. The two leishmaniasis vectors analyzed *P. duboscqi* and *P. martini* exclusively or predominantly fed on Acacia plants belonging the family Fabaceae (S1 Table). A minor representation of food crops and herbs or grasses were implicated as sand fly feeding sources. Only 3 specimens of *S. schwetzi* had fed on the invasive weed species *P. juliflora* (Fabaceae).

## Correlation in sugars between sand flies and identified plants

Nutritional profiling detected both fructose and sucrose as important sugars present in the leaves of the four identified sand fly host plants (*V. tortilis*, *S. laeta*, *V. nilotica*, *P. juliflora*) (Table 3), although only the former sugar type was present in the gut of sand flies (Table 4; Fig 3). The fructose peaks (5.02 min) were present in the plants in the same ratio as in the sand fly and sucrose was detected at 6.23 min (Fig 3).

**Table 2. Variable success rates in amplifying and sequencing plant DNA in different sandfly species.**

| Sand fly species | N = fructose positive | No. amplified (No. sequenced successfully) |
|---|---|---|
| *P. duboscqi* | 7 | 5(5) |
| *P. martini* | 12 | 10(4) |
| *S. africana africana* | 16 | 6(1) |
| *S. antennata* | 44 | 10(2) |
| *S. schwetzi* | 134 | 50(27) |
| *S. squamipleuris* | 8 | 2(1) |
| **Overall** | **221** | **83(40)** |

N = number of sandflies from which plant DNA were extracted.

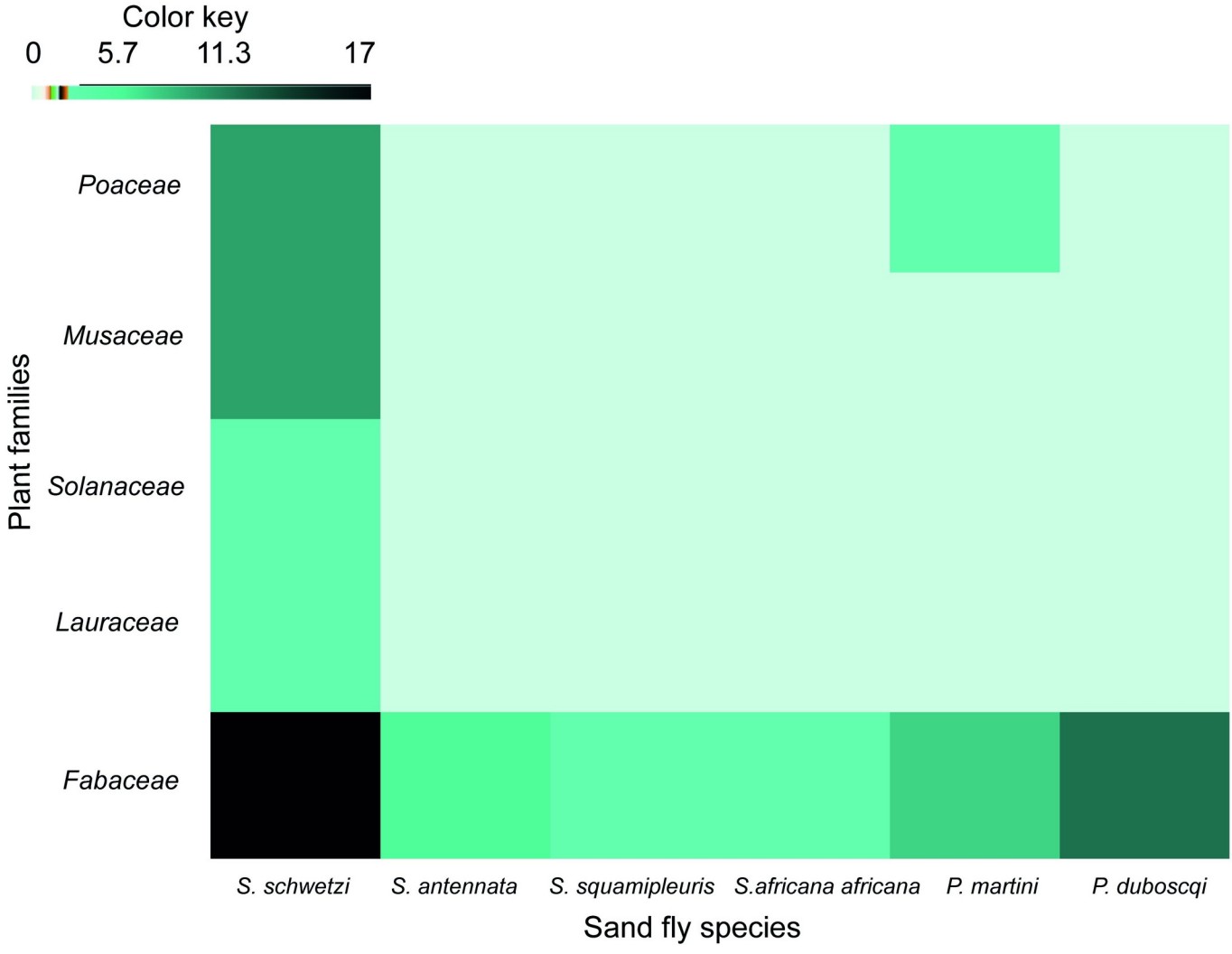

**Fig 2.** Heatmap depicting the host plant families identified for sand fly species from Rabai village, Barongo County, Kenya.

Interestingly, sucrose was absent in the leaves of the two Acacia species *S. senegal* and *V. elatior* which were not among the plants that DNA was detected in sand flies. Fructose content in the leaves significantly varied among the plants examined (F = 5.899, df = 5, P = 0.01). Fructose

**Table 3. Type and amount (ng/mg of sample) of sugars identified in the leaves of selected host plant species in the Fabaceae family.**

| Plant species | Fructose (mean±se) | Sucrose (mean±se) |
|---|---|---|
| *Vachellia tortilis* | 7.7 ± 0.9 a | 1.4 ± 0.8 ns |
| *Senegalia laeta* | 10.2 ± 3.1 ab | 5.4 ± 3.1 ns |
| *Vachellia nilotica* | 17.3 ± 2.8 abc | 6.5 ± 2.8 ns |
| *Prosopis juliflura* | 20.5 ± 2.1 bc | 9.4 ± 5.4 ns |
| *Vachellia elatior* | 17.7 ± 3.5 abc | not detected |
| *Senegalia senegal* | 23.5 ± 1.6 c | not detected |

se, standard error; values followed by same letter within a column are not significantly different at 95% level of significance; ns, not significant.

**Table 4. Type and amount (ng/mg of sample) of sugars identified in the gut of field collected sand flies.**

|  | Fructose content (mean±se) | |
| --- | --- | --- |
| **Species** | **Male** | **Female** |
| *P. martini* | - | 1.8 ± 0.3 ns |
| *S. schwetzi* | 3.0 ± 0.9 | 2.5 ± 0.5 ns |
| *S. clydei* | 3.4 ± 0.9 | 2.5 ± 0.6 ns |
| *S. antennata* | 1.4 ± 0.6 | 1.7 ± 0.3 ns |

ns, not significant; se, standard error; -, no data.

content was highest in *S. senegal* (3-fold) and *P. juliflora* (2-fold) and lowest in *V. tortilis* (0.3-fold). Fructose content was significantly higher for *S. senegal* and *P. juliflora* compared to *V. tortilis*, and between *S. senegal* and *S. laeta* (Table 3). The sucrose content in *P. juliflora* and *V. nilotica* was not significantly different from the other four plants ($F = 0.61$, $df = 5$, $P = 0.63$) (Table 3). Fructose was detected in both females and males and the amounts were not significantly different between the female sand fly species (Table 4).

## Sand fly host plants vary in their volatile profiles

Analysis of volatiles collected from four of the identified sand fly host plants (*V. tortilis*, *S. laeta*, *V. nilotica*, *P. juliflora*) by GC-MS, detected a total of 26 VOCs during the night and 23

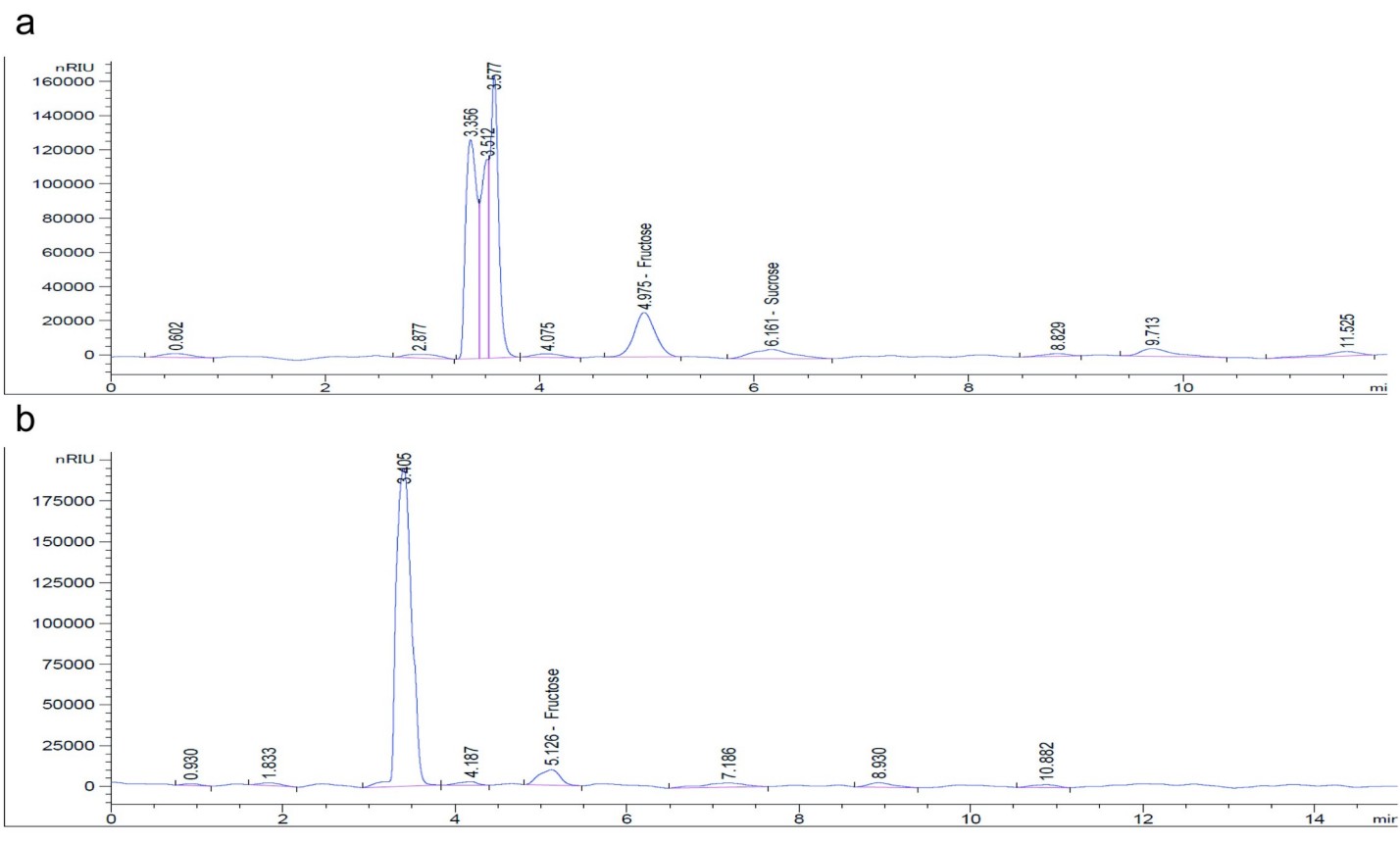

**Fig 3. HPLC profile of the sugars detected in (a) plant and (b) gut of 50 sand flies.** The peaks show names of sugars.

**Table 5. Summary of identified volatile organic compounds (VOCs) from four host plants of phelobtomine sand flies in their natural habitats in Rabai village, Baringo County, Kenya.**

| Retention time (min) | Compound name | Night collection | | | | Day collection | | | | Functional group | RI[a] | RI[b]L |
|---|---|---|---|---|---|---|---|---|---|---|---|---|
| | | *Prosopis juliflora* | *Vachellia tortilis* | *Senegalia laeta* | *Vachellia nilotica* | *Prosopis juliflora* | *Vachellia tortilis* | *Senegalia laeta* | *Vachellia nilotica* | | | |
| 6.52 | Hexanal | + | + | + | + | - | + | + | - | Aldehyde | 805 | 801 |
| 9.14 | Heptanal | - | + | + | + | + | + | + | + | Aldehyde | 905 | 907 |
| 9.82 | α-Pinene | + | + | - | - | - | - | - | - | Monoterpene | 960 | 932 |
| 10.40 | Benzaldehyde | + | + | + | + | + | + | + | + | Benzenoid | 1007 | 965 |
| 10.67 | Sabinene | + | - | - | - | + | - | - | - | Monoterpene | 1029 | 1017 |
| 10.73 | β-Pinene | + | + | + | + | + | + | + | + | Monoterpene | 1033 | 1008 |
| 10.81 | 1-Octen-3-ol | + | + | + | + | + | + | + | + | Alcohol | 1040 | 1456 |
| 10.97 | 6-methyl-5-hepten-2-one | + | + | + | + | - | + | + | + | Ketone | 1053 | 987 |
| 11.27 | Octanal | + | + | + | + | - | + | + | + | Aldehyde | 1006 | 1009 |
| 11.91 | Benzyl alcohol | + | + | + | + | + | + | + | + | Benzenoid | 1061 | 1031 |
| 12.02 | Octanol | + | - | - | - | - | + | - | - | Alcohol | 1013 | 1561 |
| 12.11 | (E)-β-Ocimene | + | + | + | + | + | + | + | + | Monoterpene | 1021 | 1044 |
| 12.45 | Acetophenone | + | + | + | + | + | + | + | + | Benzenoid | 1051 | 1076 |
| 12.57 | m-Cresol | + | + | + | - | + | + | - | - | Benzenoid | 1062 | 1098 |
| 12.64 | p-Cresol | + | + | + | + | + | + | + | + | Benzenoid | 1068 | 1077 |
| 12.85 | (Z)-linalool oxide (furanoid) | + | - | + | + | + | + | + | + | Monoterpene | 1087 | 1068 |
| 13.08 | Nonanal | + | + | + | + | + | + | + | + | Aldehyde | 1107 | 1087 |
| 13.36 | p-Cymene | + | - | + | + | + | + | + | - | Monoterpene | 1132 | 1014 |
| 14.15 | (Z)-Linalool oxide (pyranoid) | + | + | + | - | - | - | - | - | Monoterpene | 1160 | 1095 |
| 14.57 | Methyl salicylate | + | + | - | + | + | + | + | + | Benzenoid | 1198 | 1190 |
| 14.68 | Decanal | + | + | + | + | + | + | + | + | Aldehyde | 1209 | 1203 |
| 16.03 | Indole | - | - | - | + | - | - | + | + | Benzenoid | 1299 | 1298 |
| 17.66 | Longifolene | + | + | + | + | + | + | + | - | Sesquiterpene | 1425 | 1406 |
| 17.74 | α- Cedrene | + | + | + | + | + | + | + | + | Sesquiterpene | 1433 | 1413 |
| 19.89 | Phytol | - | - | + | - | - | - | - | - | Monoterpene | 1603 | 2223 |
| 20.12 | epi-Cedrol | + | - | + | + | + | - | - | - | Sesquiterpene | 1625 | 1611 |

(RT) = retention times.

RI[a] = Retention index relative to C8-C23 n- alkanes of a HP-5 MS column.

RI[b]L = Retention index obtained from literature: [30–32].

(+) = compound present and (-) = compound absent.

during the day collection. Of these volatiles, 9 were commonly detected in all the plants: benzaldehyde, β-pinene, 1-octen-3-ol, benzyl alcohol, (E)-β-ocimene, acetophenone, p-cresol, nonanal and α- cedrene (Table 5). The volatiles generally belonged to 7 functional groups: aldehyde, alcohol, benzenoid, ketone, monoterpene and sesquiterpene (Table 5). For the night emissions, 23 VOCs were detected in *P. juliflora*, 19 in *V. tortilis*, 21 in *S. laeta* and 20 in *V. nilotica*. For the day emissions, 18 VOCs were detected in *P. juliflora*, 20, 19 and 16 in *V. tortilis*, *S. laeta* and *V. nilotica*, respectively (Table 5). Overall, the volatile composition of the examined plant hosts did not differ significantly between day and night (one-way ANOSIM based on Bray–Curtis dissimilarity for day (R = 0.07, p = 0.27) (Fig 4A) and the night volatile profiles (R = 0.03, p = 0.34)) (Fig 4B). Clustering of the volatile profiles of the different plants by non-

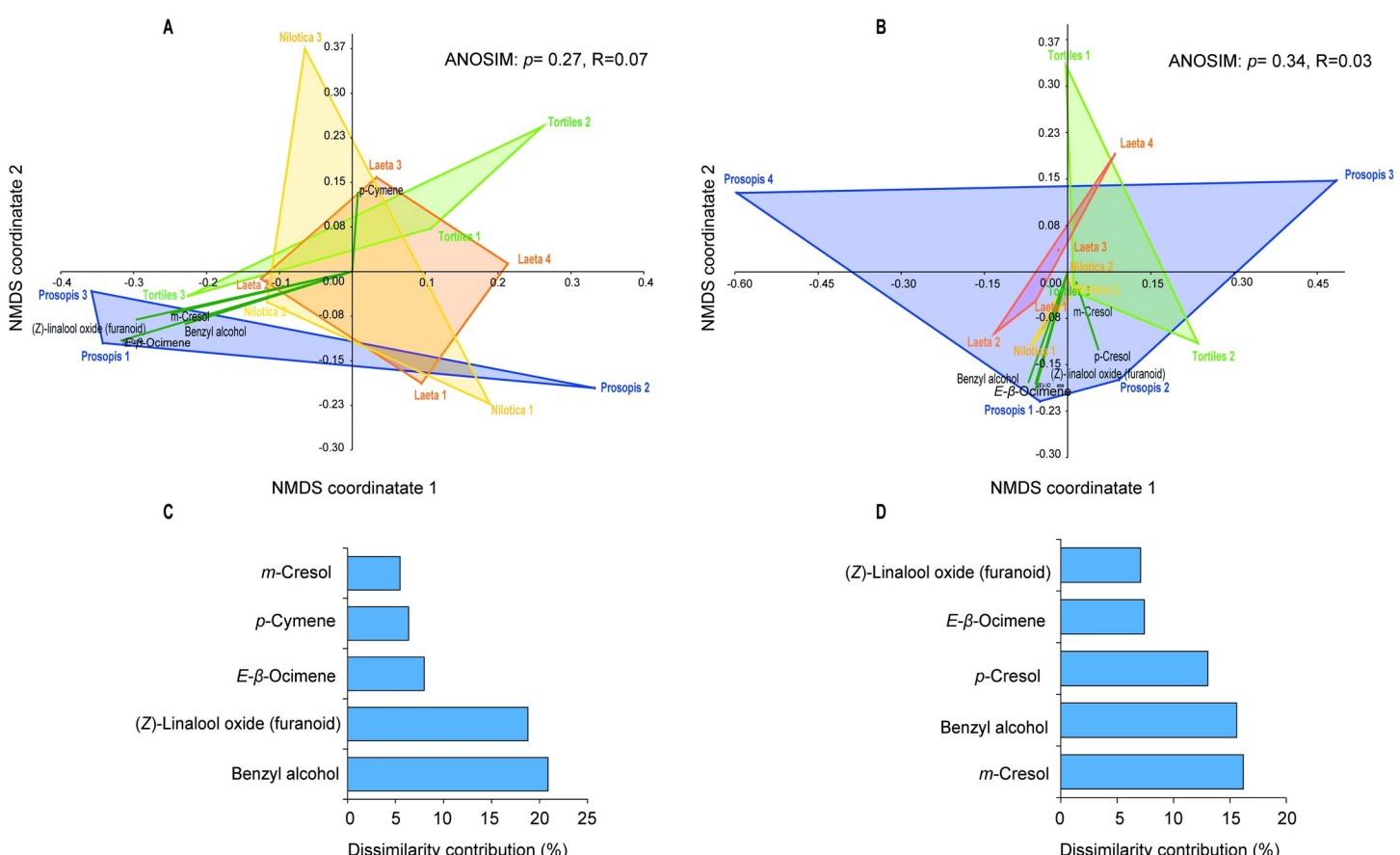

**Fig 4.** Non-metric multidimensional scaling plot (NMDS) clustering of the volatile organic compounds during the (A) day and (B) night collection. Histogram depicting the contribution of the five most important volatiles to the differentiation of all the plants types during (C) the day collection and (D) the night collection, based on Analysis of similarities (ANOSIM).

metric multidimensional scaling plot, showed that benzyl alcohol contributed more to the dissimilarity (20.9%) between the different host plant odors followed by (*Z*)-linalool oxide (furanoid) (18.8%), (*E*)-*β*-ocimene (8%), *p*-cymene (6.4%), and *m*-cresol (5%) in the day collection (Fig 4C). The order of contribution of the VOCs to the dissimilarity trends between the plants for the night volatiles were *m*-cresol (16.2%), benzyl alcohol (15.6%), *p*-cresol (13.0), (*E*)-*β*-ocimene (7.4%), and (*Z*)-linalool oxide (furanoid) (7.1%) (Fig 4D).

## Discussion

Here, we report the feeding patterns of wild-caught sand flies including *P. martini* and *P. dubosqci* vectors of visceral and cutaneous leishmaniasis, respectively, from a leishmaniasis endemic area in a dry ecological zone in Kenya. There was no sex bias in the feeding rates indicating the importance of this trait in the biology of both males and females as previously observed [5,10]. Analysis of derived rbcL sequences implicated Acacia as predominant plants readily foraged on by both sand fly sexes of diverse *Phlebotomus* and *Sergentomyia* species. Preference for feeding on plants in the Fabaceae family to which Acacia plants belong, has been documented in sand flies [14], and the malaria vector *Anopheles gambiae* [35]. However, both extent and possibly choice may be influenced by season as demonstrated in other mosquito species [36].

The vegetation cover in the study area has little undergrowth mainly with Acacia plants and weeds. Whether proclivity on Acacia is related to its biological traits, that is, a tree with a large biomass containing a range of nutrients, which may present it as a more important available and accessible food source than many small bushes or grasses or weeds to sandflies is unclear. More research is needed to unravel the strong association with Acacia. Interestingly, we found the presence of other plants in the area during the trapping period that could potentially serve as feeding sources based on succulent stems or leaves. These included *Opuntia stricta*, *Kalanchoe lanceolate*, *Euphorbia gossypina*, *Agave americana*, *Indigofera volkensii*, *Aloe secundiflora*, *Maerua subcordata*, *Caralluma russeliana* and *Balanites aegyptiaca*. Moreover, the invasive plant *P. juliflora* is rapidly spreading in the area and displacing the indigenous species such as Acacias. Yet, we found only three specimens that had fed on this plant based on analyzed rbcL sequences. This suggests that other factors may explain the seeming strong association between Acacia and diverse sand fly species at the study site during the study period. Acacia plants must possess unique adaptive qualities to explain the observed utilization not only by sand flies as observed in our study but also mosquito species [35]. Previously, high abundance of *P. orientalis* in *Acacia seyal* vegetation were attributed to factors such as the tree density and soil type and other microclimatic factors either operating independently or in combination to provide a suitable habitat for the vector reviewed in [16], as well as nutrition.

Like mosquitoes, sand fly herbivory has an underlying nutrient benefit but also a semiochemical basis. Sugar feeding is of prime relevance in the transmission of *Leishmania* parasites in sand flies [37]. We compared the sugar content present in the guts of groups of males and females and that of six selected plants in the family Fabaceae (*V. tortilis*, *S. laeta*, *V. nilotica* and *P. juliflora*, *S. senegal* and *V. elatior*). Of 8 sugars tested by HPLC, only fructose was commonly detected in the sand flies as well as leaves of these plants, indicating that this sugar is a common constituent of their diet. It is unclear if sucrose presence is a discriminating factor for feeding by sand flies given that it was not detected in Acacia species absent in their plant meals. Sucrose and trehalose have previously been reported as an important constituent of sand fly diet [38]. Maltose was also absent from their gut. The sugars detected may not represent the full profile in the insect [38] given the range of sugars tested. The detection of plant DNA confirms plant tissue feeding which sand flies generally imbibe by direct piercing and sucking [39,40]. Our analysis showed that, vegetative tissue may serve as a source of sugar, given the similarity in the detection of fructose between tissue extract and sand fly gut contents. Thus, specific sugar profiles in sand flies likely reflect those of the sources they commonly imbibe. The findings indicate that sand flies readily feed on the plant tissue to obtain these nutrients. *Vachellia tortilis* was the predominant Acacia in which plant DNA was detected in sand flies. However, this plant had the lowest content of fructose and sucrose (Table 3) although worth noting that these profiles may be dynamic. Manda et al [41], found that the preferred plants for the malaria vector *An. gambiae*, had the most sugars; this is in contrast to our findings given the sampling period. The variation in sugar types and amounts between plants suggest varied nutrient benefits and could potentially have phenotypic impact on sand fly bionomics including longevity and egg laying as observed for mosquitoes [42].

There are indications that besides sugars, Acacias are generally rich in tannins which have been found to be particularly high in their leaves and fruits and occurring in varying amounts in different species [43]. Influence of tannins on the growth and fecundity of some insects and microbe interaction has previously been reported [44]. Further assessment of this specific sand fly-Acacia interaction in relation to pathogen transmission could be a useful area for further research.

Despite knowledge of sand fly plant association, there have been minimal studies conducted to examine the semiochemical basis. Evidence from mosquito vectors have demonstrated the

major roles played by olfactory cues emitted by suitable host plants that attract vectors for feeding [9,45–47]. Examining the volatile profiles of selected identified plants in the Fabaceae family via GC-MS showed differences in VOC compositions between the plant species collected during the night- and day-time (Fig 4). Of the identified discriminating volatile compounds (benzyl alcohol, (*Z*)-linalool oxide (furanoid), (*E*)-β-ocimene, *p*-cymene, and *p/m*-cresol), (*E*)-*β*-ocimene has been implicated as signature cues for plant host location in diverse mosquito species [12]. Linalool oxide, a typical plant volatile, is known as an attractant for the mosquitoes *Aedes aegypti* [48,49] and *An. gambiae* senso lato [50,51]. The communicative function of these compounds in sand flies is yet to be established. Other compounds present in the volatile emissions of these plants including the alcohols 1-octen-3-ol, octanol and ketone 6-methyl-5-hepten-2-one have been found to elicit behavioral activity in sand flies either in the laboratory or field setting [52–54]. However, these compounds present as complex mixtures in differing amounts and ratios could interact to influence sand fly attraction to a given plant species. Electrophysiological and behavioral assays should be conducted on these VOCs to identify the attractants that can be used in monitoring sand fly populations.

Widely distributed in East Africa, sand flies are important vectors of diseases such as leishmaniasis where increased frequency of outbreaks has been reported in recent times [1]. This is a trend reflected in part by inadequacies of current chemotherapeutic approaches to curb the disease spread. The profile of sand fly vectored pathogens includes phleboviruses as demonstrated elsewhere and in Kenya recently [55–57] and there are no vaccines for the diseases they cause. The present findings provide clues towards the development of new approaches including use of the attractants in bait technologies such as ATSBs for sand fly control.

While the data generated suggest selective sand fly feeding, the success rate of plant DNA detection including sequencing from fructose positive sandflies was low (6.7%). Similarly, low rates using the same rbcL gene marker were reported by Abbasi et al [15] and Lima et al [14] analyzing wild-caught sand flies. The reasons for this observation could be ascribed to enzymatic degradation of DNA in the gut of sand flies or sugar feeding from other sources including secretions from aphids and coccids, that lack DNA [14,40,58]. More sensitive high-throughput methodologies such as next generation sequencing [13], could improve the profile of plant sequences.

## Conclusion

Our study demonstrates that Afrotropical phlebotomine sand flies including the leishmaniasis vectors *P. duboscqi* and *P. martini* of both sexes predominantly feed on plants in the Fabaceae family in a dry ecology, of which Acacia are predominantly represented. Only fructose was commonly detected in the gut of sand flies and plant vegetative parts, likely indicating that the specific sugar profile in sand flies reflects those of the plant sources they commonly feed on (Fabaceae). Future studies on the role VOCs play in plant feeding should greatly enhance our understanding of the olfactory system of sand flies and their use for management purposes.

## Supporting information

**S1 Fig. Collection of headspace volatiles from sand fly host plants in the field in Rabai village, Marigat sub-county, Kenya.**
(TIF)

**S1 Table. The composition of host plants identified from sand fly species in Rabai village, Baringo County, Kenya.**
(DOCX)

## Acknowledgments

We are grateful for the technical support of Caroline Kungu, Jackson M. Muema, Dr. Joseph Gichuhi and Mr. James Kabii. We would also like to acknowledge the help and support received from Dr. Fathiya M. Khamis, Souleymane Diallo and Simon Kiplimo.

## Author Contributions

**Conceptualization:** Baldwyn Torto, David P. Tchouassi.

**Data curation:** Iman B. Hassaballa.

**Formal analysis:** Iman B. Hassaballa, Xavier Cheseto, David P. Tchouassi.

**Funding acquisition:** Baldwyn Torto, David P. Tchouassi.

**Investigation:** Iman B. Hassaballa.

**Methodology:** Iman B. Hassaballa, Xavier Cheseto, Baldwyn Torto, David P. Tchouassi.

**Resources:** Baldwyn Torto, David P. Tchouassi.

**Supervision:** Catherine L. Sole, Baldwyn Torto, David P. Tchouassi.

**Validation:** Catherine L. Sole, Baldwyn Torto, David P. Tchouassi.

**Writing – original draft:** Iman B. Hassaballa, Baldwyn Torto, David P. Tchouassi.

**Writing – review & editing:** Iman B. Hassaballa, Catherine L. Sole, Xavier Cheseto, Baldwyn Torto, David P. Tchouassi.

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
