## [Decision Letter · Decision Letter 0]

7 Dec 2020

Dear Dr Tchouassi,

Thank you very much for submitting your manuscript "Afrotropical sand fly-host plant interactions in a leishmaniasis endemic area, Kenya" for consideration at PLOS Neglected Tropical Diseases. As with all papers reviewed by the journal, your manuscript was reviewed by members of the editorial board and by several independent reviewers. The reviewers appreciated the attention to an important topic. Based on the reviews, we are likely to accept this manuscript for publication, providing that you modify the manuscript according to the review recommendations. 

Sincerely,

Shan Lv, Ph.D.

Deputy Editor

Shan Lv

Deputy Editor

Reviewer's Responses to Questions

**Key Review Criteria Required for Acceptance?**

**Methods**

-Are the objectives of the study clearly articulated with a clear testable hypothesis stated?

-Is the study design appropriate to address the stated objectives?

-Is the population clearly described and appropriate for the hypothesis being tested?

-Is the sample size sufficient to ensure adequate power to address the hypothesis being tested?

-Were correct statistical analysis used to support conclusions?

-Are there concerns about ethical or regulatory requirements being met?

Reviewer #1: The methods were clearly articulated and included sufficient detail for reproducing the experiments. The implied hypothesis is that sand flies are selective in their acquisition of dietary sugar sources. Given the outcome that Vachellia tortillis was the preferred sugar source, even though it had the lowest content, it would be good to add something in the study about the relative abundance of the different plants in the study area.

Reviewer #2: -yes

-yes

-a little small but none the less indicative

-a little small but none the less indicative (allowances must be afforded for the inherent difficulties in collecting and procesing large numbers of insects in the middle of the African bush)

- I am not qualified to ascertain statistical data 

-none

Reviewer #3: The study is well delineated with the objectives very clear and the methodology appropriated to test the hypothesis. I did not detect problems with the statistical analysis or other techniques carried out. No concern related to ethical or regulatory requirements.

**Results**

-Does the analysis presented match the analysis plan?

-Are the results clearly and completely presented?

-Are the figures (Tables, Images) of sufficient quality for clarity?

Reviewer #1: The results are clearly presented and mostly easy to understand. However, in Table 4 under the Fructose column, it would be helpful to define what 'a', 'ab', 'bc' etc mean at the bottom of the table.

Reviewer #2: - N/A

- yes (some comments in the attached MS)

- In my opinion tables may be condensed and the full tables relegated to supplementary data 

- OK

Reviewer #3: The results are clearly presented. Only the Fig 4 A and B do not present a good resolution.

**Conclusions**

-Are the conclusions supported by the data presented?

-Are the limitations of analysis clearly described?

-Do the authors discuss how these data can be helpful to advance our understanding of the topic under study?

-Is public health relevance addressed?

Reviewer #1: The conclusions are appropriately supported by the data and analysis. Public health relevance is clearly acknowledged in the introduction and discussion. However, this sentence at line numbers 496 - 497 is a bit unclear: "The profile of sand fly vectored pathogens includes phleboviruses as demonstrated elsewhere and in Kenya recently [54–56] and without vaccines." Are you implying here that there are no vaccines for protection against phleboviruses? or that in this part of Kenya, people are not vaccinated? I think this sentence could be restated for clarity.

Reviewer #2: - yes

- I dont know

- yes

-yes

Reviewer #3: The conclusions are supported by the results and the authors discuss possible bias of the study. The investigation of basic biology of sand flies present relevance for public health, since the control of these insects need different and new approaches.

**Editorial and Data Presentation Modifications?**

Reviewer #1: A few minor edits:

Line 93 - "Male sand flies are exclusively feed on plants." Remove the word "are"

Line 120 - add a period at the end of this sentence.

Line 134 - consider changing "icipe" to "International Centre for Insect Physiology and Ecology (ICIPE)" and then use ICIPE throughout the rest of the document.

Line 187 - change "Blanstn" to "BLASTn"

Line 235 - remove the 'a' from "to generate a linear calibration curves"

Line 291 - you need a period after the word "Aldrich"

Line 447 - Use the same referencing convention used in the rest of the paper [##]

Line 460 - "likely reflects" should be "likely reflect"

Line 482 - remove the 'a' from "implicated as a signature cues"

Line 484 - what does the 'sl' mean in "Anopheles gambiae sl"?

Line 490 - "behaviroal" should be "behavioral"

Reviewer #2: see attached ms

Reviewer #3: (No Response)

**Summary and General Comments**

Reviewer #1: This is a well-designed study with clear public health significance. I found it easy to follow and enjoyable to read. The combination of analyses, including fructose positivity, DNA extractions and sequencing, and analyzing volatile organic compounds has furthered our understanding of the feeding behavior of this important vector. I think the study could be strengthened by including a survey of the relative abundance of the plant species found in the study site.

Reviewer #2: Good paper, nice work, worth publishing with some attention to detail and corrections as suggested in the annotated ms attached

Reviewer #3: Materials and Methods

132: How many CDC traps by habitats?

134: Please, for the first time to cite the Institute: use the abbreviation after the entire name: International Centre of Insect Physiology and Ecology (ICIPE). In the next times (lines 193 and 211) use only the abbreviation in capital letters (ICIPE).

214: 20 mg of each of the ground plant samples (leaves) were extracted

224: 10 μl of each plant and sand fly sample extract were analyzed

Results

Considering the technique used to detect the sand fly host plants, is it possible to identify more than one plant in the gut of one individual? If the answer is positive, this event did not happen with the samples? I suggest a phrase related to this issue. 

321: “Fructose positive rate of 44.7% (67/150) for females and 32.7% (49/150) for males significantly differ for collections indoors (χ2 = 2.8, df = 1, P < 0.09).” Considering the p value < 0.09, this is not statistically different. 

Table 3. Is there any relevance for presenting the Sample ID and Accession? If not, I suggest that you remove both columns. I am curious to know which one of those specimens were collected indoor or outdoor. I suppose you could add this information in this Table.

Table 3. You should mention in text what happened with the 7 individuals that were not identified. Were they damaged? Also the legend N/A should be changed for (-): not identified (or damaged)

Tables 4 and 5: Please, verify the legends; ng/mg is it correct? 

Fig 4: The “non-metric multidimensional scaling plot (NMDS) clustering of the volatile

organic compounds during the (A) day and (B) night collection” are not with a good resolution. Is it possible to improve the quality of the image?

Moreover, they are not related in the text (results section). Only C and D are cited in the text. 

Discussion

I suggest one paragraph about the leishmaniasis vectors species found in this study.

430: acacia – Acacia (Please, revise the document and change all by capital letters).

445: “Other adaptations than abundance may explain the seeming strong association between acacias and diverse sand fly species at the study site during the study period.” This phrase is too vague and also it is missing the reference. 

502: “While the data generated suggest selective sand fly feeding, the success rate of plant DNA detection including sequencing from fructose positive sandflies was low (<6%).” How did you find this percentage? Because if you considered 40 sequenced successfully by the total: 600, the result is 6,7%.

PLOS authors have the option to publish the peer review history of their article (what does this mean?). If published, this will include your full peer review and any attached files.

Reviewer #1: Yes: Mark F. Olson

Reviewer #2: No

Reviewer #3: No
---

## [Editor Report · Decision Letter 1]

20 Dec 2020

Dear Dr Tchouassi,

We are pleased to inform you that your manuscript 'Afrotropical sand fly-host plant interactions in a leishmaniasis endemic area, Kenya' has been provisionally accepted for publication in PLOS Neglected Tropical Diseases.

Best regards,

Shan Lv, Ph.D.

Deputy Editor

Shan Lv

Deputy Editor

---

## [Editor Report · Acceptance letter]

2 Feb 2021

Dear Dr. Tchouassi,

We are delighted to inform you that your manuscript, "Afrotropical sand fly-host plant interactions in a leishmaniasis endemic area, Kenya," has been formally accepted for publication in PLOS Neglected Tropical Diseases.

Best regards,

Shaden Kamhawi

co-Editor-in-Chief

Paul Brindley

co-Editor-in-Chief
